# Long Term Survival after Cytoreductive Surgery Combined with Perioperative Chemotherapy in Gastric Cancer Patients with Peritoneal Metastasis

**DOI:** 10.3390/cancers12010116

**Published:** 2020-01-01

**Authors:** Yutaka Yonemura, Aruna Prabhu, Shouzou Sako, Haruaki Ishibashi, Akiyoshi Mizumoto, Nobuyuki Takao, Masumi Ichinose, Shunsuke Motoi, Yang Liu, Kazurou Nishihara, Andreas Brandl, Sachio Fushida

**Affiliations:** 1NPO to Support Peritoneal Surface Malignancy Treatment, Japanese/Asian School of Peritoneal Surface Oncology, Kyouto 600-8189, Japan; dranuprabhu.surgeon@gmail.com (A.P.); sako-rgb.009@cap.ocn.ne.jp (S.S.); uranos.sanremi@i.softbank.jp (K.N.); 2Department of Regional Cancer Therapy, Peritoneal Surface Malignancy Center, Kishiwada Tokushukai Hospital, Kishiwada 596-8522, Japan; yonemu2402@gmail.com (H.I.); lymikeleo@hotmail.com (Y.L.); 3Department of Regional Cancer Therapy, Peritoneal Surface Malignancy Center, Kusatsu General Hospital, Shiga 525-8585, Japan; mizumotoakiyoshi1206@yahoo.co.jp (A.M.); nt421500@gmail.com (N.T.); ichinose1967@hotmail.co.jp (M.I.); motoi@kusatsu-gh.or.jp (S.M.); 4Digestive Unit, Champalimaud Foundation, 1400-038 Lisbon, Portugal; andreas.brandl@fundacaochampalimaud.pt; 5Department of Surgery, Kanazawa University, Kanazawa 920-8641, Japan; fushida@staff.kanazawa-u.ac.jp

**Keywords:** Cytoreductive surgery, long term survival, peritoneal metastasis, gastric cancer, neoadjuvant chemotherapy

## Abstract

The present study demonstrated prognostic factors for long-term survival in patients after a comprehensive treatment (CHT) for peritoneal metastasis (PM) from gastric cancer (GC). Materials and Methods: Among 419 patients treated with neoadjuvant intraperitoneal/systemic chemotherapy (NIPS), 266 (63.5%) patients received complete resection (CC-0) of the macroscopic tumors. In total, 184 (43.9%) patients were treated with postoperative systemic chemotherapy. Results: All patients treated who received incomplete cytoreduction (CC-1) died of GC within 6 years. In contrast, 10- year survival rates (-YSR) of CC-0 resection were 8.3% with median survival time (MST) of 20.5 months. Post-NIPS peritoneal cancer index (PCI) ≤11, and pre-NIPS PCI ≤13 were the significant favorable prognostic factors. Patients with numbers of involved peritoneal sectors ≤5 survived significant longer than those with ≥6. Both negative pre- and post-NIPS cytology was associated with significant favorable prognosis. Multivariate analyses identified pre-PCI (≤13 vs. ≥14), and cytology after NIPS (negative cytology vs. positive cytology) as independent prognostic factors. Ten year-survivors were found in patients with involvement of the greater omentum (9%), pelvic peritoneum (3%), para-colic gutter (13.9%), upper jejunum (5.6%), lower jejunum (5.5%), spermatic cord (21.9%), rectum (9.5%), ureter (6.3%), ovary (6.7%), and diaphragm (7.0%) at the time of cytoreduction. Twenty-one patients survived longer than 5 years, and 17 patients are still alive without recurrence. Conclusions: GC-PM should be removed aggressively, in patients with PCI after NIPS ≤11, PCI before NIPS ≤13, mall bowel PCI ≤2, and complete cytoreduction should be performed for metastasis in ≤5 peritoneal sectors.

## 1. Introduction

Despite the recent development of new chemotherapeutic drugs and molecular targeted drugs, the results of systemic chemotherapy for peritoneal metastasis (PM) from gastric cancer (GC) patients remain very poor [1,2,3]. Hong et al. reported that median overall survival of GC patients with PM treated with systemic chemotherapy alone was 12.5 months (95% CI; 9.4–15.5 months) and all patients died within 5 years [4].

In 1999, the Peritoneal Surface Oncology Group International (PSOGI) proposed a novel therapeutic approach, combining cytoreductive surgery and perioperative chemotherapy (POC) for patients with PM [5]. The treatment is a comprehensive treatment, and its purpose is to cure patients with PM by use of a combination of complete resection of visible metastasis and POC to eradicate micrometastatic disease. Yonemura and Glehen reported that the overall median survival time (MST) after the treatment ranged from 9.5 to 20.5 months, and 5-year survival rates were 18% and 13% [6,7]. Long term survival was significantly better after the treatment than after systemic chemotherapy alone [4,6,7].

So far, however, no study has reported 10-year survival, and the survivals of patients with special reference to the spread of PMs and the metastatic involvement of peritoneal sectors and organs after the comprehensive treatment. The present study analyzed long-term survival in patients with GC-PM after comprehensive treatment to identify prognostic factors affecting survival and to demonstrate the efficacy of the removal of the peritoneum and organs infiltrated by PM. The aim of this study was to evaluate long-term survival according to the different peritoneal sectors and organs removed.

## 2. Materials and Methods

This retrospective observational study, included in total 419 patients with GC-PM, who were treated with preoperative intraperitoneal/systemic chemotherapy (NIPS), cytoreductive surgery (CRS), and hyperthermic intraperitoneal chemoperfusion (HIPEC) between 2006 and 2019 at our institute. Inclusion criteria were pathologically proven adenocarcinoma of the stomach and pathologically proven peritoneal metastasis or positive cytology. All patients were treated with NIPS. The number of primary and recurrent cases were 281, and 138, respectively

A peritoneal port system (Hickman Subcutaneous port; BARD, Salt Lake City, UT, USA) was introduced into the abdominal cavity, and the port tip was placed in the cul-de-sac. A series of 3-week cycles of NIPS was performed. Specifically, S1 was administered orally twice daily at a dose of 60 mg/m^2^/d for 14 consecutive days, followed by 7 days of rest. Docetaxel and cisplatin were administered intraperitoneally (i.p.) at a dose of 30 mg/m^2^ each on day 1. Docetaxel and cisplatin were diluted in 500 mL of normal saline and administered through the peritoneal port system. The same doses of docetaxel and cisplatin were administered intravenously (IV) on day 8 after standard premedication. The treatment course was repeated every 3 weeks for 3 courses [8]. Before NIPS, 164 patients underwent exploratory laparoscopy, and peritoneal cancer index (PCI) was calculated and peritoneal cytology was studied.

Four to six weeks after the last cycle of preoperative chemotherapy, patients received a laparotomy beginning with incision from the xiphoid process to the pubic bone. After extensive intraperitoneal lavage using 10 L of saline, the PCI of each patient was scored [9]. Small bowel PCI (SB-PCI) is an index of total lesion size in 4 parts of the small bowel and its mesentery; it ranges from 0 to 12. Then standard peritonectomy procedures and visceral resection were performed according to the extent of disease from an oncological point of view [4]. All procedures were performed with the goal of obtaining completeness of cytoreduction (CC)-0 resection [10,11]. Following complete CRS, 255 patients were treated with HIPEC with mitomycin C (MMC) at a dose of 12.5 mg/m^2^ and cisplatin (CDDP) at 50 mg/m^2^ in 4 L of saline at an intraperitoneal temperature between 42 °C and 43.5 °C.

Removed organs and peritoneum were histologically studied by hematoxylin-eosin staining. The pathological changes related to growth of the primary tumors and peritoneal lesions were observed by two pathologists. The basic pathological changes were classified according to Japanese Classification of Gastric Carcinoma [12]. The grade ranged from 0 to 3, with 0 indicating no evidence of effect; 1a, viable tumor cells occupy more than 2/3 of the tumor area; 1b, viable tumor cells occupy more than 1/3 but less than 2/3 of the tumor area; 2, viable tumor cells remain in less than 1/3 of the tumor area; grade 3, no viable tumor cells remain. In the study, grade 0 and 1a were considered histological non-responders, and grade 1b, 2, and 3 were considered as responders. All tumors were macroscopically classified according to the Bormann classification.

After discharge, patients were followed up every 2 months until 3 years and every 6 months after 4-years as long as possible.

During follow up at the outpatient clinic, 184 patients were treated with postoperative systemic chemotherapy (40 regimens).

### 2.1. Ethical Standards

All subjects gave their informed consent for inclusion before they participated in the study. The study was conducted in accordance with the Declaration of Helsinki, and the protocol was approved by the Ethics Committee of Kishiwada Tokushukai and Kusatsu General Hospital (“A study of the safety and efficacy of a comprehensive treatment for the treatment of peritoneal metastasis from gastrointestinal cancer, H-19: Project identification code, on 26 October 2012”). All patients were informed about the adverse events of the procedure and gave their written informed consents to participate.

### 2.2. Evaluation of Complications

Postoperative complications were graded according to the system of classification reported by Dindo and colleagues [13].

### 2.3. Data Analysis

The survival analysis was performed using the Kaplan–Meier method and compared using the log rank test. Categorical variables were compared by X^2^ analysis or Fischer’s exact test. For multivariate analysis, a Cox regression model using factors which are significantly associated with death by univariate analyses was performed. Statistical analyses were performed using SPSS version 11.5 (SPSS Inc., Chicago, IL, USA). Confidence interval were calculated at the level of 95% and a *p* < 0.05 was considered significant.

## 3. Results

After CRS, in total 26 patients (6.2%) developed Clavien-Dindo Grade 3, 10 patients (2.4%) Grade 4 complications, and 6 patients (1.4%) died during the postoperative course.

Among 419 patients, 266 (63.5%) patients received complete resection (CC-0) of macroscopic diseases. The overall survival curve is shown in Figure 1, and 5- and 10- year survival rates (-YSR) were 9.6% and 5.0%.

Survival was significantly better after CC-0 resection than after incomplete cytoreduction (X^2^ = 49.3, *p* < 0.0001), and all patients treated with incomplete cytoreduction (CC-1) died of disease within 6 years with MST of 12.0 months. In contrast, 5- and 10-YSR after CC-0 resection were 14.3%, and 8.3% with MST of 20.5 months (Table 1).

The highest PCI before and after NIPS in patient surviving longer than 10 years without recurrence was 13 and 11, respectively. The MST, 5-YSR, and 10-YSR were 20.5 months, 12.8% and 7.4%, respectively, for patients with post-NIPS PCI ≤11, and 22.8 months, 18.0% and 14.0%, respectively, for patients with pre-NIPS PCI ≤13 were (Table 1).

Regarding the post-NIPS SB-PCI, patients with SB-PCI ≥3 had significantly shorter survival than those with post-NIPS SB-PCI ≤2 (*p* < 0.0001). In contrast, MST, 5-YSR and 10-YSR of patients with pre-NIPS SB-PCI ≤2 were 21.6 months, 14.9% and 8.6% (Table 1). However, there was no significant survival difference in patients between the pre-NIPS SB-PCI ≤2 and ≥3 groups).

Regarding the cytologic status before NIPS, patients with negative cytology survived significantly longer than those with positive cytology (*p* = 0.006, X^2^ = 7.54). After NIPS, the survival difference was even more significant. Patients with negative cytology after NIPS survived significantly longer than those with positive cytology after NIPS (*p* < 0.0001, X^2^ = 26.2).

Patients with positive cytology but no macroscopic PM (P0/Cy1), had a significantly better survival rate than patients with macroscopic PM (*p* = 0.0007, X^2^ = 11.4).

Histologic responders showed significant better survival than non-responders (*p* < 0.0001), X^2^ = 34.3) (Table 1)

Regarding the numbers of involved peritoneal sectors. When the number of involved peritoneal sector was ≤5 and ≥6, the MST, 5-YSR, and 10-YSR were, 21.0/11.4 months, 13.9%/1%, and 8.4%/0%, respectively (*p* < 0.0001, X^2^ = 55.5).

Lymph node metastasis (pN0/pN1 vs. pN2/pN3) was also a prognostic factor (Table 1, *p* = 0.0031, X^2^ = 8.7).

However, gender, age (<65 vs. ≥65), HIPEC (performed vs. not performed), and pre-SB-PCI (≤2 and ≥3) were not significant prognostic factor.

Multivariate analyses using Cox proportional hazard model shows that pre-PCI (≤13 vs. ≥14), and cytology after NIPS (negative cytology vs. positive cytology) emerged as independent prognostic factors with relative risk of 1.789, and 2.631, respectively (Table 2).

Table 3 shows the MST and survivals after complete removal of PM from each involved peritoneal sector and organ at the time of cytoreductive surgery.

Ten year-survivors were found in patients with involvement of the greater omentum (9%), pelvic peritoneum (3%), para-colic gutter (13.9%), upper jejunum (5.6%), lower jejunum (5.5%), spermatic cord (21.9%), rectum (9.5%), ureter (6.3%), ovary (6.7%), and diaphragm (7.0%). However, no patient with involvement in the ileum, and abdominal wall survived 10 years after the treatment. In contrast, all patients with involvement in pancreas, and ureter died of disease within 7 and 4 years after treatment (Table 3).

Twenty-one patients survived longer than 5 years, and 15 patients are still alive without recurrence (Table 4 and Table 5). All these patients received complete cytoreduction. Age of 21 5-year survivors ranged from 23 to 72 years old, and male and female were 13 and 8, respectively. Macroscopic types of primary tumors were 13 for type 4, 7 for type 3, and 1 for type 0 (Table 4). Among these 21 patients, 7 have positive peritoneal cytology without macroscopic PM (P0Cy1) before NIPS, but post-NIPS PCI of these patients changed from 0 to 1 in 3 P0/Cy1 patients. Pre- and post-NIPS PCI ranged from 0 to 13, and 0 to 10, respectively. Pre-NIPS cytological status was positive in 13 patients, but only 2 patients showed positive peritoneal cytological status after. These 2 patients are still alive without recurrence 8 and 11 years after treatments.

Seven (29.2%) of 24 P0Cy1 patients survived longer than 5 years after treatment, and 2 patients died of recurrence 5 years after treatment. One patient received oophorectomy 4.5 years after the treatment due to ovarian recurrence and died of retroperitoneal recurrence after 10 years (Table 4 and Table 5). Among 395 patients with macroscopic PM (P1), 14 (3.5%) patients survived longer than 5 years and 4 patients died of recurrence in one in the pancreas, 1 in the bone, 1 in the peritoneum, and 1 in the retroperitoneum.

## 4. Discussion

The present study clearly demonstrated that complete cytoreduction by peritonectomy procedures is considered to be essential for long-term survival [14,15,16]. MST, 5-YSR, and 10-YSR after complete cytoreduction were 20.5 months, 14.3%, and 8.3%, In contrast, all patients who received incomplete cytoreduction died within 7 years after treatment and MST, 5-YSR, and 10-YSR after incomplete cytoreduction were 12.0 months, 1.8%, and 0%, and the results were similar to those after systemic chemotherapy alone [4]. Accordingly, patients who are supposed to receive incomplete cytoreduction by exploratory laparoscopy and/or preoperative image diagnostic modalities, are not indicated for CRS, because of the high incidence of postoperative mortality and morbidity [8,9,11].

It is known that complete cytoreduction is one key element for increased oncologic outcome of patients with PM from gastric cancer. While interpreting our results, where we could not demonstrate a significant difference between patients treated with CC-0 compared to CC-1, we had to analyze the previous publications. Glehen et al. confirmed the completeness of cytoreduction score as a significant factor in multivariate analysis of their cohort [8]. In 2014, our group demonstrated the importance of completeness of cytoreduction as a significant factor in multivariate analysis comparing patients treated with CC-0 and CC-1 with CC-2 and CC-3 [11] The present study demonstrated that the PCI cutoff levels before NIPS are the more powerful prognosticator than completeness of cytoreduction. Prognosis of patients after cytoreduction could be strongly associated with the residual microscopic tumor burden [11,17]. These results may indicate that residual microscopic tumor burden in patients with PCI before NIPS higher than the cutoff level must be higher than those in patient with PCI lower than cutoff level even after complete cytoreduction. Therefore, it can be concluded that complete macroscopic cytoreduction is essential, but that patients with residual microscopic burden higher than the level that cannot be completely eradicated by intraoperative HIPEC and postoperative chemotherapy does not significantly lead to an improved overall survival even after CC-0 cytoreduction.

Regarding the prognostic factors except CCR score, the present study demonstrated that pre- PCI ≥14, post-PCI ≥12, post-SB-PCI ≥3, histologic non-responder status, lymph node involvement of pN2/pN3, positive cytology before and after NIPS, macroscopic PM, and number of involved peritoneal sectors ≥5 were significant factors indicating poor prognosis even after complete cytoreduction. Among these factors, multivariate analyses identified pre-PCI ≤13, and negative cytology after NIPS as powerful prognostic factors. MST and 10-year survival rate of patients with pre-PCI ≤13 were 22.8 months and 14.0%, and those with negative cytology after NIPS were 16.8 months, and 6.8%. These results suggest that the micrometastasis burden left in the residual peritoneum which could be completely eradicated by HIPEC and/or postoperative chemotherapy in patients with pre-PCI ≥14, post-PCI ≥12 or positive cytological, may be higher than that in patients with pre-PCI ≤13, post-PCI ≤12, or negative cytologic status.

Recently, Passot et al. reported the 10-year survival of GC patients with PM treated with CRS plus HIPEC [14]. In 127 patients, the 5- and 10 -YSR were 14% and 10% with two 10-year survivors without recurrence. They concluded that the combination of CRS and HIPEC may cure around 10% of GC-patients with PM. Our study results were almost the same as those of Passot et al. [14]. About 90% of patients who received comprehensive treatment died of recurrence in peritoneal cavity [15,16,18,19,20]. Bando et al. reported that the 5-YSR of P0CY1 patients after D2 gastrectomy alone was 2% [17], and that the cause of recurrence must be due to invisible micrometastasis burden left after gastrectomy + D2 dissection. However, the prognosis of these patients was reported to be improved by gastrectomy with perioperative chemotherapy. Yamamoto et al. treated 10 P0Cy1 patients with neoadjuvant systemic chemotherapy (NAC) with S1 plus docetaxel and CDDP plus gastrectomy [21]. Cytology became negative in 8 of 10 patients treated with systemic NAC, and survival was better in patients with negative cytology after NAC than in patients with positive cytology. Survival improvement was reported in P0Cy1 patients treated with gastrectomy plus postoperative chemotherapy using S1 [21,22,23]. NIPS using S1 plus palitaxel or docetaxel showed a trend toward improvement in overall survival [24,25,26]. Additionally, simple washing with a high volume of saline (extensive intraperitoneal lavage: EIPL) may improve the survival of P0Cy1 patients [27].

These results indicate that a certain volume of micrometastases left after CRS in the peritoneal cavity could be completely eradicated by perioperative chemotherapy. Accordingly, micrometastasis in the peritoneal cavity should be reduced as much as possible before CRS using NIPS. Compared to systemic chemotherapy, NIPS is a more powerful killer of tumor cells in intraperitoneal micrometastasis [28]. The micrometastases located outside of the surgically resected area must be eradicated just after CRS by HIPEC and early postoperative intraperitoneal chemotherapy.

According to Bando, serum Carcinoembryonic antigen (CEA) levels were significantly higher in P0CY1 patients who had recurrence after curative gastrectomy than in those who did not have recurrence. These results strongly suggest that patients with recurrence after CRS must have larger residual tumor burden which could be completely eradicated by intraoperative and postoperative chemotherapy.

To date, no study has reported the long-term survival of patients after comprehensive treatment with special reference to PM involving peritoneal sectors and abdominal organs. The purpose of the present study was to clarify the effect of removing infiltrating peritoneal sectors and organs on postoperative survival. Long-term survival could be obtained after surgical removal of PM-involved areas including 13 peritoneal sectors and abdominal organs except pancreas capsule, or ureter. The 10-YSR of patients with involvement of paracolic gutter, spermatic cord or sac, rectum, uterus, ovary, upper jejunum, and lower jejunum were 13.9%, 21.9%, 9.5%, 6.3%, 6.7%, 5.5%, and 5.6%, respectively. Patients with metastasis of diaphragmatic peritoneum and ileum who were disease-free for more than 5-years did not survive 10 years after treatment.

If the PM is found on the peritoneal surface or organs except pancreas and ureter, and the pre-PCI ≤ 14, post PCI ≤ 11, post SB-PCI ≤ 2 and number of involved peritoneal sector ≤5 after NIPS are confirmed, CCR-0 resection should be performed. Some patients with non-response after NIPS, and positive cytology after NIPS survived longer than 10 years but their survival rates were significantly lower than those with response to NIPS, and negative cytology after NIPS.

One of the limitations of this study is that it only included patients treated by one surgical group in two institutions, where patients were selected, and oncologic treatment plans recommended on the basis of our long experience in peritoneal surface malignancies. In this setting, selection bias was inevitably a factor in patient enrollment. The most important factors in patient selection were PCI, tumor response and biology, and performance status. Furthermore, due to the nature of a retrospective analysis, the treatment plan for some patients differed, including factors such as pretreatment PCI, the usage of HIPEC, and postoperative chemotherapy.

## 5. Conclusions

From the present data, PM should be removed aggressively to achieve complete cytoreduction for PM, when the patients have PCI after NIPS ≤ 11, PCI before NIPS ≤ 13, SB-PCI ≤ 2, involvement of ≤5 peritoneal sectors.

Only one randomized controlled study confirmed the effects of CRS plus POC [29]. New methods to more completely eradicate the residual tumor burden after CRS should be developed in the future.

## Figures and Tables

**Figure 1 cancers-12-00116-f001:**
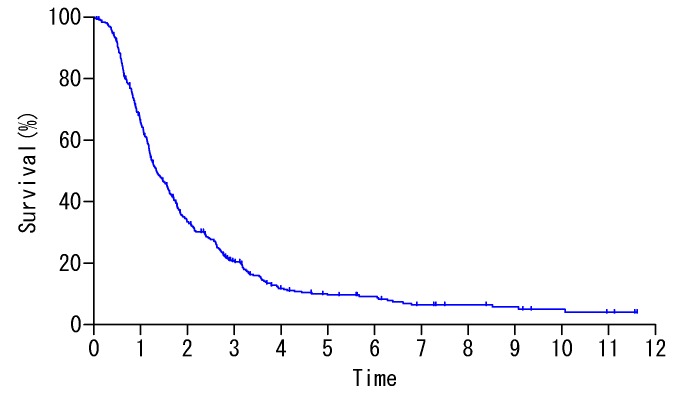
Overall survival of 419 patients.

**Table 1 cancers-12-00116-t001:** Median survival time, 5- and 10-year survival rates after treatment.

Clinico-Pathological Parameters	Classification	No. of Patients	MST (Months)	5-YSR	10-YSR	*p*, X^2^
Completeness of cytoreduction	CC-0	266	20.5	14.3%	8.3%	<0.0001
	CC-1	153	12	1.8%	0%	49.3
PCI after NIPS	≤11	292	20.5	12.8%	7.4%	<0.0001
	≥12	117	9.6	2.2%	0%	55.7
PCI before NIPS	≤13	129	22.8	18%	14%	<0.0001
	≥14	35	13.7	0%	0%	14.49
Small bowel PCI after NIPS	≤2	241	21.6	14.9%	8.6%	<0.0001
	≥3	178	11.8	2.4%	NR	34.3
Small bowel PCI before NIPS	≤2	123	27.6	18.6%	14.9%	NS
	≥3	43	24.2	0%	0%	
Histological effcts	Non-responders	280	14,2	5.6%	0%	<0.0001
	Responders	139	23.8	15.8%	9.4%	34.3
Lymph node metastasis	pN0, pN1	141	19	11.4%	7.2%	0.0031
	pN2, pN3	278	14	7.8%	1.7%	8.7
Histologic type	Differentiated	35	22.7	34.6%	14.8%	0.0127
	Poorly	384	15.4	8.2%	4.40%	6.2
Cytology before NIPS	Negative	146	21.1	12.20%	3.90%	0.006
	Positive	273	14.3	8%	6.60%	7.54
Cytology after NIPS	Negative	293	16.8	11.50%	6.50%	<0.0001
	Positive	98	11.9	3.50%	2.30%	26.2
Peritoneal/Cytologic status	Macroscopic PM	395	15.4	8.60%	3.60%	0.0007
	P0Cy1	24	31.2	24.75	24.75	11.4
No. of involved peritoneal sectors	≤5	252	21	13.90%	8.40%	<0.0001
	≥6	167	11.4	1%	0%	55.49

MST: median survival time; -YSR: -year survival rate; PCI: Peritoneal Cancer Index; NIPS: neoadjuvant intraperitoneal/systemic chemotherapy; PM: peritoneal metastasis; P0Cy1: no peritoneal metastasis but positive cytology. CC: Completeness of cytoreduction.

**Table 2 cancers-12-00116-t002:** Independent prognostic factors.

Prognostic Factors	Classification	X^2^	*p*	RR	CI
Completeness of cytoreduction	CC-0 vs. CC-1	0.002	NS	0.983	0.497–1.912
PCI after NIPS	≤11 vs. ≥12	0.133	NS	1.148	0.545–2.419
Small bowel PCI after NIPS	≤2 vs. ≥3	0.941	NS	1.331	0.747–2.375
PCI before NIPS	≤13 vs. ≥14	5.189	0.023	1.789	1.085–2.953
Histological effects	no responder vs. responder	2.041	NS	0.702	0.433–1.140
Lymph node metastasis	pN0, pN1 vs. pN2, pN3	2.481	NS	1.358	0.978–1.989
Histologic type	Differentiated vs. poorly	0.669	NS	1.305	0.669–2.542
Cytology before NIPS	Negative vs. positive	0.7	NS	0.822	0.519–1.300
Cytology after NIPS	Negative vs. positive	10.637	0.001	2.631	0.471–4.705
Peritoneal/Cytologic status	Macroscopic PM vs. P0Cy1	0.367	NS	0.807	0.405–1.611
No. of involved peritoneal sectors	≤5 vs. ≥6	0.37	NS	1.336	0.526–3.393

PCI: Peritoneal Cancer Index; CC: Completeness of cytoreduction; PM: peritoneal metastasis; NIPS: neoadjuvant intraperitoneal/systemic chemotherapy; P0Cy1: no peritoneal metastasis but positive cytology.

**Table 3 cancers-12-00116-t003:** Median survival time, and 5- and 10-year survival rates of patients with metastasis on the peritoneal sectors and organs after comprehensive treatment.

Site of Metastatic Involvement	No. of Patients with Metastatic Involvement	MST	5-YSR	10-YSR
Right diaphragm	29	13.1	8%	7%
Left diaphragm	30	12	11.9%	7%
Greater omentum	113	16.4	10.8%	9%
Pelvic peritoneum	107	17	6.1%	3%
Para-colic gutter	31	14.5	13.9%	13.9%
Upper jejunum	45	18	5.6%	5.6%
Lower jejunum	37	19.9	5.5%	5.5%
Upper ileum	35	16.2	7.1%	NR ^#^
Lower ileum	42	15.2	7.3%	NR ^&^
Spermatic cord	5	74	21.9%	21.9%
Rectum	57	16.4	9.5%	9.5%
Uterus	19	17	6.3%	6.3%
Ovary	33	17.8	6.7%	6,7%
Abdominal wall	17	17.4	6.9%	NR ^$^
Ureter	12	17.6	0%	0%
Pancreas	17	13.4	30.1%	0%

MST: median survival time; -YSR: -year survival rate; NR: not reached, NR ^#^; Two patients are surviving longer than 5 and 6 years without recurrence, NR ^&^; Two patients are surviving longer than 5 and 6 years, NR ^$^; One is surviving longer than 6 years after treatment.

**Table 4 cancers-12-00116-t004:** Clinical data, Peritoneal Cancer Index before NIPS (pre-PCI), histological responses of peritoneal metastasis after NIPS and cytology status before and after NIPS in 21 patients with 5-year survival.

Case No.	Age	Gender	Histologic Differentiation	Bormann Type	Pre PCI	Tumor Response after NIPS	Cytology before NIPS	Cytology after NIPS
1	55	F	poorly	4	2	MR	Positive	Positive
2	57	M	poorly	4	0(P0Cy1)	CR	Positive	negative
3	71	M	poorly	4	2	PR	Positive	negative
4	72	M	poorly	4	2	CR	Positive	negative
5	62	M	poorly	3	2	PR	Positive	negative
6	65	f	poorly	4	0(P0Cy1)	CR	Positive	negative
7	47	M	poorly	0	0(P0Cy1)	PR	Positive	negative
8	65	F	poorly	4	2	PR	negative	negative
9	64	M	moderately	3	1	NR	negative	negative
10	59	F	poorly	3	4	NR	negative	negative
11	35	F	poorly	4	4	MR	negative	negative
12	23	M	poorly	4	10	PR	Positive	Positive
13	36	F	poorly	4	0(P0Cy1)	PR	Positive	negative
14	25	F	poorly	4	0(P0Cy1)	PR	Positive	negative
15	64	M	moderately	4	2	CR	Positive	negative
16	43	F	moderately	3	0(P0Cy1)	MR	Positive	negative
17	61	M	poorly	3	13	CR	negative	negative
18	57	M	poorly	4	1	PR	Positive	negative
19	62	M	moderately	3	2	PR	negative	negative
20	60	M	moderately	3	2	PR	negative	negative
21	39	M	poorly	4	0(P0Cy1)	NC	Positive	negative

PCI: Peritoneal Cancer Index; NIPS: neoadjuvant intraperitoneal/systemic chemotherapy; M: male; F: female; P0Cy1: no peritoneal metastasis but positive cytology; CR: complete response; PR: partial response; NR: no response; MR: minor response; NC: not classified.

**Table 5 cancers-12-00116-t005:** operative details, hyperthermic intraperitoneal chemoperfusion (HIPEC), postoperative chemotherapy and outcome of 5-year survivors.

Case No.	PCI at CRS	CC	No. of Removed Peritoneal Sectors	No. of Pathologically Involved Sectors	Removed Organs	HIPEC	Postoperative Chemo.	Status at Last Follow-Up
1	2	CC-0	4	1	TG, D2, RHC, EPR, BSO, SP	Done	DCS Ip	11 Y, AWD
2	0	CC-0	0	0	TG, D2	Done	XELODA	6 Y, AWD
3	1	CC-0	2	0	DG, D2	No	S1, capecitabine	7 Y, AWD
4	2	CC-0	2	3	DG, RHC	No	no	7 Y, AWD
5	2	CC-0	3	1	TG, D2, SP, GB, RHC	No	DCS Ip	11 Y, AWD
6	1	CC-0	3	0	TG, D2, SP, GB, HYS, BSO	Done	DCS IV + PO S1	7 Y, AWD
7	0	CC-0	2	0	TG, D2,SP, GB, EPR	Done	no	112 Y, AWD
8	3	CC-0	5	3	TG, D2,SP, GB, HYS, BSO	Done	S1	8 Y, died of recurrence in pancreas
9	1	CC-0	6	1	DG, D2,GB	Done	S1	9 Y, AWD
10	4	CC-0	6	1	DG, D2, GB, HYS, BSO	Done	Immunotherapy	6 Y, died of bone mets.
11	6	CC-0	6	1	TG, D2, SP, GB, RHC, ,EPR	Done	S1	8 Y, died of peritoneal mets.
12	6	CC-0	5	3	TG, D2, RHC, EPR, SP, GB	Done	Capecitabine + CDDP + Herceptin	8 Y, AWD
13	1	CC-0	5	9	TG, D2, HYS, BSO	Done	S1, CapeOX	6 Y, died of peritoneal mets.
14	1	CC-0	4	1	TG, D2, SP, GB	Done	S1	10 Y, died of peritoneal mets
15	1	CC-0	5	1	TG, D2, SP, GB	Done	DCS Ip, capecitabine	7 Y, died of retroperitoneal recurrence
16	0	CC-0	5	0	TG, D2, SBR, SP, GB, HYS, BSO	Done	S1	7Y. AWD
17	10	CC-0	7	7	TG, D2, RHC, SBR, GB	Done	S1	6 Y, AWD
18	1	CC-0	2	1	TG, G2, SP	Done	DCS IV + nab-paclitaxel	9 Y, AWD
19	3	CC-0	5	1	DG, D2	Done	S1 + Irrinotecan	7 Y, AWD
20	1	CC-0	5	1	TG, D2, SP, GB	Done	S1	6 Y, AWD
21	0	CC-0	7	0	TG, D2, SP, GB, RHC, ,EPR	Done	S1	6 Y, AWD

No.: number; PCI: Peritoneal Cancer Index; CRS: Cytoreductive Surgery; CC: Completeness of Cytoreduction; TG: total gastrectomy; DG: distal gastrectony; RHC: right hemicolectomy; D2: D2 lymph node dissection SP: splenectomy; GB: cholecystectomy; EPR: extraperitoneal rectal resection; SBR: small bowel resection; HYS: hysterectomy; BSO: bilateral salpingo-oophorectomy; DCS: Docetaxel + Cisplatin; CapeOX: Capecitabine + Oxaliplatin; AWD: alive without disease; Y: years; IP: intraperitoneal; IV: intravenous.

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
