# Peer review of "Long Term Survival after Cytoreductive Surgery Combined with Perioperative Chemotherapy in Gastric Cancer Patients with Peritoneal Metastasis"

_cancers, 2020, doi:10.3390/cancers12010116_

Round 1

Reviewer 1 Report

The authors have revised the manuscript according to the previous queries in most part, but some concerns remain.

Because completeness of cytoreduction is a factor reported to be crucial for patient survival in other studies but not in this one, it is better to be discussed in the DISCUSSION section. Regarding to the metastatic sites as prognostic factor, I suggest to state it clearly as metastatic sites found during operation, not only “Ten year-survivors were found in patients with involvement of the greater omentum (9%), pelvic peritoneum (3%), para-colic gutter (13.9%), upper jejunum (5.6%), lower jejunum (5.5%), spermatic cord (21.9%), rectum (9.5%), ureter (6.3%), and ovary (6.7%).” Otherwise, the readers will mistake it as the finding at initial diagnosis. NR (not reached) in Table 3 is not appropriate. As the authors say, some patients are alive for more than 5 years. These patients will be censored in the 10-year survival curve. It is inappropriate to say NR for 10-YSR. To avoid confusion, I suggest to write it as XX % (number of survive / number of patients not censored). Although this is a cohort with large patient numbers, there are still some caveat to be discussed. For example, this is not a well controlled, randomized study, many important factors are not prespecified, not all patients receive exploratory laparotomy before NIPS, not all patients receive HIPEC, not all patients receive postoperative systemic chemotherapy, et al. The question in first query also represents a point to be discussed.

Author Response

Reviewer #1

The authors have revised the manuscript according to the previous queries in most part, but some concerns remain.

Because completeness of cytoreduction is a factor reported to be crucial for patient survival in other studies but not in this one, it is better to be discussed in the DISCUSSION section.

Thank you for your valuable comment. We included a new paragraph in the discussion section regarding this topic.

Regarding to the metastatic sites as prognostic factor, I suggest to state it clearly as metastatic sites found during operation, not only “Ten year-survivors were found in patients with involvement of the greater omentum (9%), pelvic peritoneum (3%), para-colic gutter (13.9%), upper jejunum (5.6%), lower jejunum (5.5%), spermatic cord (21.9%), rectum (9.5%), ureter (6.3%), and ovary (6.7%).” Otherwise, the readers will mistake it as the finding at initial diagnosis.

We added “at the time of cytoreduction” to this sentence.

 During the period of 2 months while I am revising the MS, one patients with involvement on right and left diaphragm survived 10-years after the treatment. Accordingly, 10-year survival rates of patients with  involvement on the right and left diaphragm were 7.0% and 7.0%, respectively. So, I changed 10-year survival of the two groups as 7% and 7% in Table 3.

However, two patients with involvement on upper/lower ileum are surviving 5 and 6 years after treatment. One patient with involvement on abdominal wall is surviving 6 years after treatment without recurrence.

Accordingly, I added the sentences in the annotation of Table 3, as followings; NR# ; Two patients are surviving longer than 5 and 6 years without recurrence after treatment, NR&;Two patients are surviving longer than 5 and 6 years after treatment, NR$ ; One is surviving longer than 6 years after treatment.

Although this is a cohort with large patient numbers, there are still some caveat to be discussed. For example, this is not a well controlled, randomized study, many important factors are not prespecified, not all patients receive exploratory laparotomy before NIPS, not all patients receive HIPEC, not all patients receive postoperative systemic chemotherapy, et al. The question in first query also represents a point to be discussed.

Thank you for your important comment. We added a sentence mentioning these aspects to the “limitations of this study” section at the end of the discussion.

Reviewer 2 Report

Thank you 

Author Response

Thank you

Reviewer 3 Report

Changes have sufficiently improved the paper

Author Response

Thank you

This manuscript is a resubmission of an earlier submission. The following is a list of the peer review reports and author responses from that submission.

Round 1

Reviewer 1 Report

Cancers-623894
Long term survival after cytoreductive surgery combined with perioperative chemotherapy in gastric cancer patients with peritoneal metastasis

The benefit of surgery for gastric cancer patients with peritoneal metastasis is still controversial. Previous reports suggested there probably is prolonged survival for patients receiving cytoreductive surgery, especially for those with R0 resection. In this paper, Yonemura and colleagues reported their observation on 419 gastric cancer patients who received resection of metastatic lesions after neoadjuvant intraperitoneal/systemic chemotherapy (NIPS). They analyzed the risk factors associated with survival and the major advantage of this study is the long term survival follow up period. Although the work deserves to be announced, there are issues to be addressed first.

Several prognostic factors are analyzed for their contribution to overall survival (Table 2), including completeness of cytoreduction, PCI before NIPS, PCI after NIPS, et al. However, only PCI before NIPS and cytology after NIPS remained significant in the multivariate analysis. It is interesting to see the reason why completeness of cytoreduction, a factor reported to be crucial for patients survival (as listed in ref 14,15,16), is insignificant in this patients cohort. Table 3: the authors stated that patients with involvement of the greater omentum, pelvic peritoneum, et al. had long term survival in contrast to patients with diaphragmatic peritoneum, ileum and et al. However, this is findings during operation, not the pretreatment assessment. The metastatic or cancer involved sites might disappear during the surgical assessment if they responded very well to the NIPS. Therefore, it will overlook the metastatic sites which had better response to NIPS and carried the better prognosis. In addition, NR (not reached) in Table 3 is not appropriate. Is it better to be 0%?

I suggest to delete the relevant description of the analysis.

Why only 164 patients underwent exploratory laparoscopy before NIPS? It became an significant bias in the multivariate analysis, especially this turns out an independent prognostic factors (Table 2). Some abbreviations miss their full name in first appearance. For example, SB-PCI in ABSTRACT. Multi variate should be multivariate. The statement “The present study will show that long-term survival depends on determining which peritoneal sectors and organs should be removed” in INTRODUCTION is confusing and should be rewrote. In MATERIALS AND METHODS: the enrollment period should be stated clearly instead of “during the last 13 years”. Were both docetaxel and cisplatin given at the same dose of 30 mg/m2 on day 1 (MATERIALS AND METHOLDS)? Please clarify. Was HIPEC a pre-specified procedure in the study? Why only 255 out of 419 patients received HIPEC? The fact that 184 patients were treated with postoperative systemic chemotherapy should be mentioned in the ABSTRACT. Was it a pre-specified procedure in the study? I am confused by the statement “After CC-0 resection Grade3….., and 6 (2.9%) patients” in RESULTS. In addition, why 10 means 2.5% but 6 means 2.9%? All figures and tables should be self-explainable. What does P/Cy status mean in Table 1? What does MST mean in Table 3? PCI in table 4? CCR in table 5? In line 159-162, page5: what do type 4, type 3 and other types mean? What does “No. of removed peritoneal sector” mean in Table 5? The caveat of this study should be discussed in DISCUSSION. I strongly suggest to have the whole manuscript revised by an English-editor before resubmission.

Author Response

Reviewer #1:

The benefit of surgery for gastric cancer patients with peritoneal metastasis is still ざcontroversial. Previous reports suggested there probably is prolonged survival for patients receiving cytoreductive surgery, especially for those with R0 resection. In this paper, Yonemura and colleagues reported their observation on 419 gastric cancer patients who received resection of metastatic lesions after neoadjuvant intraperitoneal/systemic chemotherapy (NIPS). They analyzed the risk factors associated with survival and the major advantage of this study is the long term survival follow up period. Although the work deserves to be announced, there are issues to be addressed first.

Several prognostic factors are analyzed for their contribution to overall survival (Table 2), including completeness of cytoreduction, PCI before NIPS, PCI after NIPS, et al. However, only PCI before NIPS and cytology after NIPS remained significant in the multivariate analysis. It is interesting to see the reason why completeness of cytoreduction, a factor reported to be crucial for patient´s survival (as listed in ref 14,15,16), is insignificant in this patient´s cohort.

RE: Thank you very much for your valuable comment. Indeed, completeness of cytoreduction was significant in univariate analysis, but lost its significance in the Cox Regression analysis. One possible explanation might be, that in all of the mentioned studies the effect of complete cytoreduction proved significance in univariate analysis only and in cross-disease publications. Additionally, the effect of patient selection obviously contributes to low CC1 rates and is linked to the pattern of peritoneal spread, as many patients with diffuse metastasis of gastric cancer are not considered for CRS. Especially in patients with gastric cancer, where the selection criteria for CRS are stricter compared to other diseases like ovarian, colorectal, or even Pseudomyxoma peritonei., the effect of the selection bias might appear stronger.

Table 3: the authors stated that patients with involvement of the greater omentum, pelvic peritoneum, et al. had long term survival in contrast to patients with diaphragmatic peritoneum, ileum and et al. However, this is findings during operation, not the pretreatment assessment. The metastatic or cancer involved sites might disappear during the surgical assessment if they responded very well to the NIPS. Therefore, it will overlook the metastatic sites which had better response to NIPS and carried the better prognosis. In addition, NR (not reached) in Table 3 is not appropriate. Is it better to be 0%?

I suggest to delete the relevant description of the analysis.

Thank you for your important comment. As the final decision of performing CRS in these patients is always taken after neoadjuvant chemotherapy, we believe the findings of the operation have definitely more impact on decision taking as the initial status. Naturally, response and other factors influence these findings, but we aim to provide assistance in taking decisions for these patients, based on this large series.

Regarding your point of using 0% instead of not reached, we believe using 0% would create the impression, that all patients died before reaching 10 years. In fact, each group (except of Ureter, and Pancreas) contains patients, who are alive for more than 5 years, but at the time of analysis have not reached the point of 10-year survival. Therefore, we suppose the usage of not reached is correct .

Why only 164 patients underwent exploratory laparoscopy before NIPS? It became an significant bias in the multivariate analysis, especially this turns out an independent prognostic factors (Table 2).

RE: The reason that 39.1% (164/419) patients underwent laparoscopy before NIPS is related to the change in practice and partially missing data. Not every patient received laparoscopy at the time of i.p. Port implantation.

We are aware of the influence of this factor, which was not present in all of the patients in the multivariate analysis, but believe as it proved to be a significant factor to be included in the analysis.

Some abbreviations miss their full name in first appearance. For example, SB-PCI in ABSTRACT. Multi variate should be multivariate. The statement “The present study will show that long-term survival depends on determining which peritoneal sectors and organs should be removed” in INTRODUCTION is confusing and should be rewrote. In MATERIALS AND METHODS: the enrollment period should be stated clearly instead of “during the last 13 years”.

RE: Thank you for mentioning these important aspects. We corrected them.

Were both docetaxel and cisplatin given at the same dose of 30 mg/m2 on day 1 (MATERIALS AND METHOLDS)? Please clarify.

RE: That is correct. We adopted the manuscript in order to state it clearer.

Was HIPEC a pre-specified procedure in the study? Why only 255 out of 419 patients received HIPEC?

RE: HIPEC was not used in every patient as it was not defined as the standard treatment in this study, and therefore up to the surgeons choice. Additionally, it was avoided in few cases of intra-operative alterations, such as excessive blood loss, inability to maintain adequate urine output or mean arterial pressure.

The fact that 184 patients were treated with postoperative systemic chemotherapy should be mentioned in the ABSTRACT. Was it a pre-specified procedure in the study?

RE: We added the information to the Abstract. The usage of postoperative systemic chemotherapy was not a pre-specified procedure. The decision was taken for each patient after individual discussion in a multi-disciplinary team meeting.  

I am confused by the statement “After CC-0 resection Grade3….., and 6 (2.9%) patients” in RESULTS. In addition, why 10 means 2.5% but 6 means 2.9%?

RE: We are sorry about this typographic error. We corrected this statement accordingly.

All figures and tables should be self-explainable. What does P/Cy status mean in Table 1? What does MST mean in Table 3? PCI in table 4? CCR in table 5?

RE: We added Table legends to each Table.

In line 159-162, page5: what do type 4, type 3 and other types mean?

RE: The macroscopic classification was performed according to Bormann. We added this information to the material and methods section.

What does “No. of removed peritoneal sector” mean in Table 5?

RE: No. means number. We added this abbreviation to the Table legend.

The caveat of this study should be discussed in DISCUSSION.

RE: As this is the world largest study for CRS in peritoneal metastasized gastric cancer, we are not sure, what you mean by caveat? Could you please clarify this statement?

I strongly suggest to have the whole manuscript revised by an English-editor before resubmission.

RE: We revised the manuscript by a native speaker.

.

Reviewer 2 Report

Thank you for asking me to review this important article showing the long term survival after CRS and Preoperative chemotherapy in gastric cancer with PM. its quite interesting now reporting 10 years survival in gastric cancer with PM However, I do have the following comments

1.aim of this study should be clearly written in the introduction section.

2.NIPS was given However the author uses abbreviation in the chemotherapy please write the type of chemotherapy clearly and avoid abbreviation 

3.what is the rate of nephrotoxicity in this protocol as cisplatin is known to cause nephrotoxicity and what is the rate of completion of this protocol

4.the author should clarify if the gastric cancer are primary or recurrent 

5.in the materials and method section line 72 the author wrote 164 patients underwent exploratory laparoscopy and PCI was calculated what happened to the rest of the patients were their PCI calculated radiologically ? please comment how accurate is the method in calculating PCI .

6.in the materials and method section line 81 authors wrote 244 patients treated with HIPEC i could not understand the right number of patients who had HIPEC and it will be helpful if a table or figure showing the exact number of patients ,how many had NIPS ? How many had HIPEC?How many were excluded 

7.in the discussion section , line 222 the author wrote about the effect of HIPEC and EPIC in micrometastses I would like the author to comments on the role of Early post operative intra peritoneal chemotherapy Post CRS in the discussion

8. There should be KM survival Curve

9.please write the limitations of your study in the discussion section 

Author Response

Reviewer #2:

Thank you for asking me to review this important article showing the long term survival after CRS and Preoperative chemotherapy in gastric cancer with PM. its quite interesting now reporting 10 years survival in gastric cancer with PM However, I do have the following comments

1.aim of this study should be clearly written in the introduction section.

RE: We added the aim of the study to the Introduction section

2.NIPS was given However the author uses abbreviation in the chemotherapy please write the type of chemotherapy clearly and avoid abbreviation

RE: Thank you for your important comment. We added this information.

3.what is the rate of nephrotoxicity in this protocol as cisplatin is known to cause nephrotoxicity and what is the rate of completion of this protocol

RE: Only two patients developed nephrotoxicity during the treatment. The low rate (1 of 194 patients) of nephrotoxicity as well as any other medical complication for the NIPS regimen in patients with gastric cancer was already published by our group (Canbay E et al. Ann Surg Oncol (2014) 21:1147–1152). These results might be easily explained by the relatively low dose of normothermic 30mg/m2 cisplatin i.p. and 50mg/m2 during HIPEC.

4.the author should clarify if the gastric cancer are primary or recurrent  

 No. of primary and recurrent cases were 281, 1nd 138, respectively. The number are added in Materials and methods

5.in the materials and method section line 72 the author wrote 164 patients underwent exploratory laparoscopy and PCI was calculated what happened to the rest of the patients were their PCI calculated radiologically? please comment how accurate is the method in calculating PCI.

RE: Thank you for your valuable comment. The reason that 39.1% (164/419) patients underwent laparoscopy before NIPS is related to the change in practice and partially missing data. Not every patient received laparoscopy at the time of i.p. Port implantation. We did not use radiologically PCI calculation, because of its rather low accuracy. Every PCI in this manuscript was evaluated surgically.

6.in the materials and method section line 81 authors wrote 244 patients treated with HIPEC i could not understand the right number of patients who had HIPEC and it will be helpful if a table or figure showing the exact number of patients ,how many had NIPS ? How many had HIPEC? How many were excluded

RE: All patients were treated with NIPS, as stated in Material and Methods (page 2, line 64). In total 255 patients were treated with HIPEC (page 2, line 81).

7.in the discussion section, line 222 the author wrote about the effect of HIPEC and EPIC in micrometastses I would like the author to comments on the role of Early post operative intra peritoneal chemotherapy Post CRS in the discussion

RE: Thank you for your constructive impulse. We did not perform EPIC in any patient. In total, 184 patients received systemic or a combination of systemic and intraperitoneal chemotherapy in an “adjuvant” setting, starting 4-6 weeks after surgery. Nevertheless EPIC, is an interesting concept, which is mainly used in patients with positive cytology or high risk for free intraabdominal tumor cells without the presence of peritoneal metastasis in Asia. As EPIC was not object of our study, we believe a discussion about this topic would be misleading and confusing.

There should be KM survival Curve

I added the over all survival curves, in Figure 1.

9.please write the limitations of your study in the discussion section

RE: Thank you for your important comment. We added a limitations section at the end of the discussion section.

 [AB1]@ Dr. Yonemura: Could you please provide this data?

 [AB2]I agree. I will create them easily, if you could send me the data.

Reviewer 3 Report

Dear authors, thanks for your work. I feel there is no clear statement about the statistical design and characteristics of this study: this is a retrospective observational study. Please improve the material and methods and statistical design part.

Table 5 is not effective and totally chaotic. Please simplify or delete it. You can’t put only abbreviation or acronyms, these should be a minority of words in the table.

Author Response

Dear authors, thanks for your work. I feel there is no clear statement about the statistical design and characteristics of this study: this is a retrospective observational study. Please improve the material and methods and statistical design part.

RE: Thank you for your comment. We enriched the material and methods part according to your suggestions.

Table 5 is not effective and totally chaotic. Please simplify or delete it. You can’t put only abbreviation or acronyms, these should be a minority of words in the table.

RE: Thank you for your valuable comment. We believe this Table is of importance, as it illustrates the treatment details of these few patients with long term survival. We added a Table legend in order to improve the readability